# Bread Surplus: A Cumulative Waste or a Staple Material for High-Value Products?

**DOI:** 10.3390/molecules27238410

**Published:** 2022-12-01

**Authors:** Ines Ben Rejeb, Ichrak Charfi, Safa Baraketi, Hanine Hached, Mohamed Gargouri

**Affiliations:** 1Department of Biological and Chemical Engineering, LR05ES08 Laboratory of Microbial Ecology and Technology, National Institute of Applied Sciences and Technology (INSAT), University of Carthage, BP 676, Tunis 1080, Tunisia; 2Laboratoire Innovation et Valorisation Pour une Industrie Alimentaire Durable, Ecole Supérieure des Industries Alimentaires de Tunis, Tunis 1003, Tunisia; 3Department of Biological Engineering, Université Libre de Tunis (ULT), Tunis 1002, Tunisia

**Keywords:** bread residues, valorization, value-added products, processing technologies

## Abstract

Food waste has been widely valorized in the past years in order to develop eco-friendly materials. Among others, bread waste is currently of increasing interest, as it is considered a huge global issue with serious environmental impacts and significant economic losses that have become even greater in the post-pandemic years due to an increase in cereal prices, which has led to higher production costs and bread prices. Owing to its richness in polysaccharides, bread waste has been previously studied for its physico-chemical characteristics and its numerous biotechnological applications. The present review highlights the re-use of bread waste and its valorization as a valuable resource by making value-added products through numerous technological processes to increase efficiency at all stages. Many research studies reporting several transformation methods of surplus bread into ethanol, lactic acid, succinic acid, biohydrogen, hydroxymethylfurfural, proteins and pigments, glucose–fructose syrup, aroma compounds, and enzymes are widely discussed. The wide variety of suggested applications for recycling bread waste provides significant insights into the role of technology development in potentially maximizing resource recovery and consequently contributing to environmental performance by reducing the amount of bread waste in landfills.

## 1. Introduction

Food waste has significantly shaped queries in the twenty-first century. According to a study by the Food and Agriculture Organization of the United Nations (FAO), food is being produced at a scale of 1.3 billion metric tons per year [1]. A UN analysis estimates that 931 million tons of food is wasted annually between families, retail stores, and the food service industry, which raises the possibility that the actual amount of food wasted globally may be more than twice the previous estimations. In fact, waste is determined by several factors along the supply chain. The UN Environment Programme (UNEP) Food Waste Index Report states that households account for 61% of food waste, food service accounts for 26%, and retail accounts for 13% [2].

Bread has been one of the highest food waste categories. Considering the fact of being a staple food, estimates have suggested that bread has been produced globally at 100 million tons per year, leading to hundreds of tons that have been wasted daily [3]. Indeed, despite innovations in storage and packaging methods, bread waste is inevitable. Due to its highly nutritious properties with various nutrients, bread is highly susceptible to staling and spoiling. In fact, bread is made from cereal grains that are quite indigestible in their raw state, and as a result, is considered resistant to microbial attack. However, due to the combined effects of heat and moisture during its baking process, starch in bread becomes digestible in its gelatinized form and therefore prone to microbial attack, which explains bread’s short shelf life [4]. Additionally, edible bread is wasted during the manufacturing process, which is explained by the production of substandard products [4]. These reasons, in addition to consumers’ preferences for freshly baked products, have led to bread being piled up from bakeries to retailers and households. 

The fact that large amounts of bread are produced but not consumed by humans has substantial negative effects on the environment and the economy. From greenhouse gas emissions to billions of dollars lost annually, bread surplus has been a challenge faced by governments, charitable organizations, corporations, and individuals while aiming to halve the amounts and the impacts of global bread waste, and food waste in general, by 2030 [5]. Indeed, the situation of bread surplus is significantly worse in developed countries, since over half of the bread produced is wasted, raising serious economic concerns. In fact, according to the FAO Food Price Index, cereal prices increased in September 2020 by 2.0% from the previous month. World wheat prices increased up to 41 percent in 2020 compared to the previous year [5]. Moreover, The Russia–Ukraine war has disrupted wheat exports, driving wheat prices up by 60%. In fact, Russia and Ukraine have collectively accounted for about 30% of global wheat exports over the last three years. Conflict-related export disruptions in both countries prompted a surge in global prices of wheat and coarse grain. The FAO Cereal Price Index was 17.1% higher in March than it was in February.

Additionally, in the past two years, maritime freight shipping costs have roughly tripled due to COVID-19 [6]. The root cause has been the lack of shipping containers due to pandemic-related transportation inefficiencies that have hit a breaking point [6]. In consequence, higher prices of these inputs will undoubtedly translate into higher production costs and higher bread prices. This will inevitably lead to serious economic losses. In addition, as an organic biogenic waste, bread waste could cause serious environmental issues and pollution of natural habitats with CO_2_ emissions [7].

To comply with environmental quality objectives set by waste management systems, solutions for limiting bread waste have progressively been established. One of the valorization routes is using bread waste as animal feed. In fact, feeding animals food scraps is one of the bases of the Food Recovery Hierarchy, and farmers have used this method for centuries. By following legislation for the selection of proper and safe food waste, bread leftovers or stale dry bread can offer a great alternative method to obtain animal feed at a much lower cost [8]. While a small amount of bread waste is being recycled as animal feedstock, the rest ends up in landfills. Hence, the valorization of waste bread has been in the spotlight in recent years to manage approximately 1.2 million tons of wasted bread per annum [4]. Meanwhile, many efforts have been initiated in the past decades to investigate methods of repurposing bread residues into fuel and chemicals such as bioethanol, biohydrogen, succinic acid, and various added-value products that can be exploited in versatile industries.

The present review aims at highlighting potential applications for recycling bread waste into valuable products through chemical and biotechnological approaches.

## 2. Search Methodology

Extensive bibliographic research was conducted using the scientific databases Web of Science, PubMed, Scopus, and annual reports from organizations (FAO and UNEP) by selecting articles and reviews from the past decades (2008–2022). The following keywords were used in the research: technology, valorization, value-added products, stale bread, surplus bread, and bread waste. To facilitate the screening, the research was refined by using the terms: bread making, ethanol, lactic acid, succinic acid, biohydrogen, hydroxymethylfurfural, proteins, aroma compounds, growth medium, and enzymes.

All the articles that met the criteria to be included in the study were analyzed. Reports obtained were evaluated by screening abstracts to discard unnecessary, incomplete, or irrelevant literature. A total of 80 articles were analyzed in full text and classified by groups based on the molecules studied, types of technologies used, and production yields.

## 3. Bread Waste: A Rising Problem or a Valuable Resource?

### 3.1. Bread Shelf Life and Reasons for Bread Staling

Bread is a universal food that is sold and consumed across the entire social and geographical spectrum. Hence, its wastage is likely to be ubiquitous. However, as with any other waste, bread waste can also have serious impacts. Therefore, understanding the process of waste generation can help in limiting the wasted amount and potentially the issues that come with it.

When bread becomes wasted food, it goes through a phase of staling. Depending on many factors, bread goes through several physical–chemical changes from the moment of its production. The staling process causes the loss of important sensory parameters of bread, such as flavor, texture, and an increase in crumb firmness as well as loss of freshness. Fadda et al. [9] thoroughly investigated the main causes of bread staling, which they summed up in their review. Firstly, their research using previous data postulated that the ingredients used in bread production could have a significant impact on its staling process. In fact, the quality of flour was mainly investigated to highlight that the type of flour or the absence of some components, essentially amylose, can improve nutritional aspects and bread aging [10,11]. They also reported the role of lipid and shortening in retarding the crumb firming process, revealing that the supplementation of new ingredients (i.e., fat monoglycerides, sodium stearoyl lactylate, and bioemulsifier) significantly reduced bread staling. Moreover, Fadda et al. [9] noted other processing factors affecting the staling rate such as baking technology, sourdough fermentation, process parameters, and storage temperature.

Furthermore, one of the important aspects is bread’s richness with many nutrients and other compounds that can make it highly perishable. The typical composition of bread is presented in Table 1.

### 3.2. Environmental and Economic Impacts of Bread Surplus

According to the 2016 data from the Food and Agriculture Organization of the UN, bread is the fourth most wasted food in the world with an estimated rate of 29.1%, causing significant environmental concerns and economic losses worldwide.

Due to the huge amounts being piled up throughout the bread life cycle (Figure 1), bread waste presents a serious environmental impact. Considering the fact of being organic biogenic waste, studies have shown that bread is responsible for gas emissions in the form of carbon dioxide or even methane. It was estimated that each 800 g loaf of bread is responsible for generating 100 L of biogas (over 60% methane, and the rest is CO_2_). Therefore, alongside many other pollutants, bread waste contributes drastically to air pollution with CO_2_ emissions. In fact, researchers have determined the carbon footprint of 800 g of bread, which ranges from 977 to 1244 g of CO_2_ [4].

Based on the data collected from 113 sales points, Lanfranchi et al. [13] determined the footprint of bread waste in Italy. Altogether, a total of 6780 kg of bread per month generated over 721 kg of CO_2_. Thus, the carbon dioxide emissions of bread waste in a year are equivalent to the emissions emitted by a runabout traveling 5996 km.

In a recent study, it was estimated that about 20 million slices of bread are thrown away in the UK daily, which leads to an annual wastage of 292,000 tons, corresponding to 584,000 tons of equivalent CO_2_ emissions [7].

Bread loss reflects also on the economical state. With a global annual production of over 100 million tons, it is estimated that over 900,000 tons annually of all manufactured bread is wasted, which is around 24 million slices of bread every day [14]. In 2015, a study in Flanders estimated that 25% of the total bread and bakery production is lost throughout the whole supply chain (Figure 2), which equals 69 million kg of bread every year. Indeed, in the Netherlands, the economic loss of bread waste accounts for over USD 464 million [15]. France also holds a great place in bread wastage, with an annual estimation of over EUR 16 billion or over USD 18 billion [16]. In Turkey, 1.8 billion loaves of bread go to waste every year, which could be translated into USD 502.25 million (TL 1.4 billion) [17].

In 2017, Lanfranchi et al. [13] conducted a survey on 113 companies located in the territory of southern Italy (Sicily), showing a high wastage of bread. They noted a total waste of 6780 kilos of bread a month, which is translated to a total loss of over USD 16,000 (about EUR 14,500) in economic terms.

Wasting bread is also a serious problem in Mediterranean Arab countries. In 2018, the Tunisian National Institute for consumption estimated that 5% of food is being wasted per year, which is equivalent to IQD 572 million (USD 197 million). In fact, a joint study by the FAO, the Ministry of Agriculture and Hydraulic Resources, and the INC [18] showed that between families, bakeries, and university restaurants, bread is the most wasted food, estimated at 15.7% at a scale of IQD 450 million in 2017 (Figure 3).

## 4. Bread Waste: Valorization into Value-Added Products

### 4.1. Reprocessing Bread for Bread Making

One of the methods used in the baking industry to valorize low-quality bread and bread returned from the distribution network is reprocessing. The recycling of non-standard bread saves natural food resources and provides economic benefits for bakers.

Recycled bread offers many economic benefits for bread producers. Despite the low quality of recycled bread compared to the ingredients, which affect its luster, brittleness, softness, taste, specific aroma, and smell, the development of resource-saving technologies was found to be relevant. In addition, inappropriate storage conditions generate fungal mycelium on the bread surface; fungi enzymes break down proteins, fats, and carbohydrates, leading to a deterioration in the bread’s properties (unpleasant appearance, smell, and taste).

Savkina et al. [19] investigated the impact of recycled old bread on bread quality and its microbiological safety. Only fresh bread within the shelf-life date is allowed to be recycled in Russia. The authors studied the substitution of flour at 15, 20, 25, 35, 50, and 100% by recycling old breadcrumbs in sourdough samples. The obtained results showed that bread dosage in the rye-dense sourdough greater than 25% negatively affected the sourdough quality without any nutrient deficiency noticed. At a recycled bread dosage of 25%, the resulting quality was comparable to traditional rye-dense sourdough, but the crumbliness was 1.5 times lower compared to the control.

Sensory evaluation showed that bread made with fermented old bread in sourdough had interesting sensory characteristics (crust shape and color, taste, odor, chewiness, and porosity) comparable to the control. Moreover, old, recycled bread did not show any significant effect on the microbial contamination (molds and yeasts) of new bread.

The Holland Company Sonextra Sustain, a special starter for reprocessing bread, has developed a new concept for the reprocessing of bread. It offers to the customer the possibility to proceed safely to produce good-quality bread. This special technology included in the starter for reprocessing bread prevents the growth of bacteria that form spores and cause mold, resulting in tastier and softer bread.

Consequently, reprocessing bread is considered an alternative way to manage waste bread, as it contributes to tastier bread with upgraded quality, lower costs, and less environmental damage.

### 4.2. Ethanol Production

Ethanol is considered one of the most promising fuel sources. It may be used as an alternative fuel instead of fossil fuels. Sustainable bioethanol is mostly produced by microbial fermentation (commonly yeast) of agricultural and/or food wastes.

Among raw materials used in fuel ethanol production, starchy materials are the most common feedstocks. One of the most promising, highly accessible, and inexpensive raw materials is bread residue. It contains a significant amount of starch, a polymer composed of glucose molecules, which is easily hydrolyzed to monomeric sugars using amylases. According to Dewettinck et al. [20], the amounts of starch and simple sugars in bread were found to be 500–750 and 3–50 g/kg, respectively. Moreover, bread waste contains 100–150 g/kg of proteins, which, after hydrolysis to peptides and amino acids, accelerate yeast growth and fermentation.

Pietrzak and Kawa-Rygielska [21] studied the direct conversion of starch in waste wheat–rye bread to ethanol using a granular starch hydrolyzing enzyme (GSHE). They suggested three raw material pretreatment methods, i.e., enzymatic prehydrolysis, microwave irradiation, and sonification, for ethanol yield and fermentation course improvement compared to separate hydrolysis and fermentation (SHF). The obtained results from the work conducted showed that all pretreatment methods increased the final ethanol yield compared to unpretreated waste bread fermentation. The fermentation of unpretreated waste bread ended with an 80.00% ethanol yield, while the use of pretreated raw material improved ethanol yield by 3–8%. Furthermore, the highest values of ethanol productivity were achieved in all studied samples within the first 48 h of fermentation.

Datta et al. detailed trials including the utilization of bread waste as the sole source to manufacture glucose using *Aspergillus niger* through solid-state fermentation, followed by the synthesis of bioethanol from glucose using *Saccharomyces cerevisiae* [22]. The solid-state fermentation of waste bread using *A. niger* produces a multienzyme solution containing amylolytic and proteolytic enzymes. The crude enzymatic extract was used for bread waste hydrolysis at 55 °C and 300 rpm, resulting in approximately 145 g/L of glucose. The hydrolysate was then used to produce ethanol at a concentration of 54 ± 2 g/L with an achieved conversion efficiency of 72%.

In the same context, Mihajlovski et al. [23] also optimized bioethanol production from bread waste as a biomass source by response surface methodology (RSM). The effect of fermentation duration (24–72 h) and waste brewer’s yeast inoculum (1–4%) on ethanol production was studied. The optimized conditions, obtained by central composite design (CCD), were 48.6 h of fermentation and 2.85% of inoculum. Under these conditions, the maximum ethanol production of 2.06% was reached. The obtained results demonstrate that the use of waste bread offers multiple benefits related to environmental protection, reduction in production costs, and saving of fossil fuels.

Interestingly, a British beer company called “Toast Ale” have also used bread and bakery wastes to produce brews. In fact, beer is produced primarily from cereal grains (mostly barley), hops, and water. Recently, small breweries have started to use surplus bread in their recipes, substituting part of the malted barley, originally used as a source of sugar for fermentation. Two recipes were investigated in this study. In both recipes, 25–28% of the original malt was substituted with dried bread. The remaining ingredients were maintained as used in the standard process of beer production. These new beers are opening up a space in the craft beer industry, where the sustainability of food waste valorization options is needed in addition to flavor [24].

The described processes are important in the sustainable chemical industry because they convert waste food into value-added products such as ethanol. The previous methods of bioconversion of waste bread into alcoholic products limit the use of chemical additives, extra distillation, and purification steps and contribute to preserving the environment and human health along with lower production costs.

### 4.3. Lactic Acid Production

Lactic acid is one of the most important organic acids that has great value with variable applications in many industries such as food, beverage, pharmaceutical, chemical, and cosmetics. It has attracted the interest of many researchers, especially in the production of biodegradable poly lactic acid (PLA), well-known as a bioplastic material [25]. According to the report of Global View Research, the global lactic acid market size reached its highest peak in 2020, and it is expected to expand in the following years with an estimated annual growth rate of 0.8% [26].

The production of lactic acid can be performed by chemical synthesis or by fermentation using starchy or sugary biomass. Thus, the richness of bread waste with starch and other nutritional compounds provides a great renewable resource for lactic acid production in both processes.

The chemical conversion of bread residues into lactic acid can be carried out via hydrothermal treatment using alkaline catalysts (NaOH, KOH, Ca(OH)_2_, LiOH, and K_2_CO_3_). As described in the study by Sánchez et al. [27], a mixture of bread and alkaline solutions in a ratio of 1/25 (solid/liquid) was heated at 300 °C with continuous stirring for 30 min. The maximum yields of lactic acid production of 38.11 ± 0.2%, 34.46 ± 0.21%, and 72.90 ± 4.45% were reached in the mixture of KOH (0.4 M), NaOH (0.6 M), and Ca(OH)_2_ (3.5 M), respectively.

Despite the time-saving advantage presented by the previous method, it was not able to effectively eliminate the environmental side effects of the production of undesirable and harmful compounds, i.e., lactic acid D-isomer [25]. Hence, researchers tend to lean towards the fermentation process due to environmental impact and full control of the product outcome. In other words, an optically pure L(+)- or D(−)-lactic acid can be obtained by microbial fermentation of renewable resources depending on the microorganism used [28]. The fermentative production of lactic acid is based on the conversion of sugary, starchy, or even lignocellulosic biomass via different strains of microorganisms. The fermentation process also has its challenges. The yield of fermentation can be significantly affected by various factors, namely, physiochemical factors (pH, temperature, nutrients, substrate concentrations, etc.), the microorganisms, and the type of biomass used [29].

Furthermore, Pleissner et al. [30] experimentally investigated the fermentation method using restaurant food waste as starchy feedstuff. This process was based on simultaneous saccharification and fermentation through two different strains (*Streptococcus* sp. and *Lactobacillus* sp.). Maximum productivity of 2.16 g L^−1^ h^−1^ was observed in *Streptococcus* sp. strain, yielding 58 g/L of lactic acid from a 20% (*w*/*w*) food waste blend.

Recently, Hassan et al. [31] introduced another process of fermentation using a combination of kitchen food waste and banana peels in a ratio of 3/1 (*w*/*w*) using *Enterococcus durans* BP130. Although this process was conducted under harsh conditions, it delivered interesting results: maximum productivity of 0.24 g L^−1^ h^−1^ yielding the highest lactic acid concentration of 28.8 g/L.

### 4.4. Succinic Acid Production

Succinic acid (SA, C_4_H_6_O_4_) is a dicarboxylic acid that has attracted great interest worldwide. The presence of two carboxyl groups makes SA a precursor molecule for the synthesis of many chemical compounds. Hence, it has a wide range of applications in industries such as pharmaceuticals, food, polymers, plasticizers, and green solvents [32].

Originally, succinic acid was produced via a fossil-based system through catalytic hydrogenation of petrochemical maleic anhydride. However, this process can be expensive and harmful to the environment [33].

Another route for SA synthesis is bio-based production via fermentation from renewable resources. This process has attracted research interest because it provides lower energy consumption due to milder operation conditions and lower dependence on a single feedstock. The common microorganisms used in fermentative SA bio-production are *Actinobacillus succinogenes*, *Anaerobiospirillum succiniciproducens*, *Mannheimia succiniciproducens,* and recombinant *Escherichia coli* [34]. Most of these microorganisms use glucose as a carbon source for SA production. Thus, due to its richness of carbohydrates and other nutrients, bread waste was found to be an excellent substrate for SA biosynthesis.

In general, the fermentation process is carried out through two main steps. Firstly, the processing of bread waste, via hydrolysis using solid-state fermentations of *Aspergillus awamori* and *Aspergillus oryzae,* produces enzyme complexes rich in amylolytic and proteolytic enzymes, respectively. As described in the study of Leung et al. [35], this step generates a hydrolysate containing over 100 g/L glucose and 490 mg/L free amino nitrogen. Moreover, the bread hydrolysate is then used as the sole feedstock for *Actinobacillus succinogenes* fermentation, leading to the production of 47.3 g/L succinic acid with a yield and productivity of 1.16 g SA/g glucose and 1.12 g/L h, respectively. Overall, 0.55 g of succinic acid per gram of bread was obtained via this process [35]. The biochemical pathway for SA production from glucose (carbon source) by *A. succinogenes* is described by Gadkari et al. [32] as follows: PEP/Pyruvate carboxylase route: 0.5 C_6_H_12_O_6_ + NADH + H^+^ + CO_2_ → C_4_H_6_O_4_ + H_2_O + NAD^+^
PEP carboxy kinase route: 0.5 C_6_H_12_O_6_ + CO_2_ + NADH + H^+^ + ADP + Pi → C_4_H_6_O_4_ + H_2_O + NAD^+^ + ATP

Following these experimental demonstrations, Zhang et al. [36] conducted the same fermentation process by using cake and pastry wastes separately as feedstocks, producing SA at 24.8 and 31.7 g/L with a yield of 0.80 and 0.67 g SA/g sugar, respectively. According to this study, the SA productivity achieved with cake and pastry wastes was 0.79 and 0.87 g L^−1^ h^−1^, with an overall yield of 0.28 and 0.35 g succinic acid per gram of substrate, respectively.

### 4.5. Biohydrogen Production

Hydrogen is considered a valuable component due to its possible use as a promising energy source in the future because it is clean and renewable. Hence, many studies were conducted to find the best production process, such as steam reforming, electrolysis, biophotolysis of water, and fermentation. Most researchers seem to prefer the fermentation technique for H_2_ generation because it is eco-friendly and requires less external energy [37].

Bread waste is an excellent biomass for biohydrogen (H_2_) production due to its wide range of nutrients.

The process of biohydrogen production can be divided into two steps. Firstly, the processing of bread waste biomass enables the breakdown of all nutrients as macromolecules (starch and protein) into monomers (glucose and free amino nitrogen). As mentioned in the research of Han et al. [38], the hydrolysis of this biomass can be performed via crude enzymes, which were generated by two microorganisms: *Aspergillus awamori* and *Aspergillus oryzae* via solid-state fermentation. In the second step, the waste bread hydrolysate was then used for biohydrogen production by anaerobic sludge in a continuously stirred tank reactor (CSTR). This was the first study that reported continuous biohydrogen production from waste bread by anaerobic sludge with a yield of 109.5 mL hydrogen/g of waste bread at chemical oxygen demand (COD) concentration of 6000 mg/L [38].

Recent research has thoroughly investigated other processes such as dark fermentation using a wide range of substrates including agricultural and industrial starchy wastes with a low level of undesirable compounds. This process was theoretically shown in the past to be a fruitful process for hydrogen production with mild operation [39].

Recently, Jung et al. [40] showed the feasibility of dark fermentation for biohydrogen production. They stated that conducting the process of dark fermentation from food waste using hybrid immobilization in mesophilic conditions yielded a hydrogen production rate (HPR) of 9.82 ± 0.30 L/L-d at an organic loading rate (OLR) of 74.7 g hexose/L-d, leading to a yield of 1.25 ± 0.04 mol H_2_/mol hexose_consumed_ [40]. However, this framework solely pertains to a laboratory scale, and the scaling-up option is yet to be tested.

### 4.6. Hydroxymethylfurfural (HMF) Synthesis

Bread waste offers renewable biomass that can be exploited in hydroxymethylfurfural (HMF) production. Also known as 5-(hydroxymethyl)-2-furancarboxaldehyde and 5-(hydroxymethyl)-2-furaldehyde, HMF displays a wide range of applications in versatile industries due to its capacity to be upgraded into many chemicals such as medicines, polymers, resins, fungicides, and biofuels [41].

Fundamentally, the synthesis of HMF from food waste involves three reactions: hydrolysis of glucan to glucose, isomerization of glucose to fructose, and dehydration of fructose to HMF.

Previous research by Yu et al. [42] showed that HMF production from bread waste is based on the thermochemical process using a mixture of polar aprotic solvents (DMSO, THF, ACN, or acetone) and water in a ratio of 1/1 (*v*/*v*) under heating at 140 °C with SnCl_4_ as a catalyst. Under these conditions, a maximum HMF yield of 26–27 mol% was achieved in the mixtures of ACN/H_2_O, acetone/H_2_O, and DMSO/H_2_O. However, this method fails to eliminate any undesirable side reactions (rehydration of HMF to levulinic acid and polymerization of HMF to humins), which can significantly affect the production yield [42].

The applicability of these results was then tested in the research of Cao et al. [43,44] on starchy food waste (i.e., bread, rice, and spaghetti), showing a yield much higher than the previous study. They began by testing two different biochars as a catalyst followed by optimizing the reaction conditions to heating at 180 °C for 20 min in a mixture of dimethylsulfoxide (DMSO)/deionized water (DW) in a ratio of 3:1 (*v*/*v*). Under these optimum conditions and by using sulfonated and acid-activated wood biochars, the yield of the conversion was found to be improved to reach 30.4 Cmol% [43] and 30.2 Cmol% [44] respectively. This study expanded the understanding of the effects of biochar catalysts on desirable and undesirable reactions, showing higher catalytic activity of biochars towards starch hydrolysis and fructose dehydration.

### 4.7. Bread Waste for Protein and Pigment Production

One of the main value-added products of waste bread is proteins and pigments, such as carotenoids, which are present in photosynthetic microorganisms and plants. They are tetraterpene pigments that display orange, yellow, red, and purple colors [45]. These isoprenoid pigments are used as nutraceuticals and health additives and have been recently developed to be fit for pharmaceutical uses to prevent many diseases, including cancer. They are known for their antioxidant characteristics, which confer the ability to protect cells against oxidative damage by neutralizing free radicals [46].

Carotenoids are also used as natural pigments in the food industry as additives to conserve the shelf life of many food products or to enhance organoleptic features [47] and are used in many other industrial areas as well, such as paper and textile industries.

Gmoser et al. [48] used waste bread as a substrate for the edible filamentous fungus *Neurospora intermedia* in a double-staged fermentation process combining submerged and solid-state fermentations to produce pigments such as carotenoids. Submerged fermentation was performed initially on thin stillage to produce the fungal biomass *N. Intermedia.* The latter was then used for solid-state fermentation in the presence of air on waste bread to generate a feed rich in carotenoids and proteins. The obtained results showed that the production of carotenoids was noticeably higher than that obtained using inoculation with a spore solution. Furthermore, proteins showed an increase of approximately 161% compared to the bread waste before the fermentation process.

Another study conducted by Haque et al. [49] investigated the production of natural pigments (orange, yellow, and red) applied in food and textile industries. The authors used recovered amino acids and sugars from bakery waste hydrolysate as substrates for the filamentous fungus *Monascus purpureus*, as well as bakery waste, to produce proteins (enzymes) such as glucoamylase and protease. The bakery waste hydrolysate was generated using different species of filamentous fungi *Aspergillus awamori* and *Aspergillus oryzae* and was then used as a substrate for *M. purpureus* in submerged fermentation. The same fungus that used bakery waste as a substrate in solid-state fermentation was investigated to produce the enzymes glucoamylase and protease. The highest yield of pigment, which presented around 24 AU units/g glucose, was obtained with the bakery hydrolysate with a low initial glucose concentration of 5 g/L.

Moreover, glucoamylase and protease enzyme activities were 8 U/g at 55% of moisture and 117 U/g at 60% of moisture, respectively, at 30 °C [49].

### 4.8. Bioconversion of Waste Bread to Glucose–Fructose Syrup

Many have studied green and sustainable methods to convert waste bread into fermentable sugars, as it is mainly composed of starch (natural polymer consisting of polysaccharides).

In fact, the bioconversion of waste bread into glucose–fructose syrup was investigated by Riaukaite et al. [50], involving enzymatic hydrolysis to produce glucose via a two-step process using amylolytic enzymes and then isomerization to produce fructose.

The first step of the enzymatic hydrolysis process is the liquefaction of bread slurry to produce oligosaccharides, such as dextrin, using an α-amylase enzyme that breaks down long chains of polysaccharides making up the starch into shorter chains [51].The second step of this process is using the glucoamylase enzyme to break down dextrin into monosaccharides—actual glucose molecules [52].

Then, fructose is produced by enzymatic isomerization of the glucose molecules using the glucose isomerase enzyme. High yields of the final glucose–fructose syrup were found to depend on the amount of bread and enzymes used [50]. In fact, bioconversion of a minimal amount of waste bread resulted in low yields of glucose. Additionally, a high amount of waste bread induced a highly viscous slurry, halting enzyme activity in breaking down starch [53]. According to Riaukaite et al. [50], the produced glucose–fructose syrup was composed of 45.27 ± 0.55% glucose and 40.32 ± 0.80% fructose.

Another study conducted by Haque et al. [54] investigated the bioconversion of beverage waste to fructose syrup as a low-cost value-added product in a multi-step process that involves enzymatic hydrolysis, activated carbon treatment, ion-exchange chromatography, and ligand-exchange chromatography. As a result, 47.5% of sugars present in the hydrolysate were recovered as a syrup rich in fructose, while the remaining sugars were in the glucose-rich stream.

### 4.9. Production of Aroma Compounds

An aroma compound, also known as an odorant, fragrance, or flavor, is a chemical compound characterized by a smell or odor when two conditions are established. On one hand, the compound must be volatile. This helps it to easily be detected by the olfactory system in the upper part of the nose. On the other hand, a sufficiently high concentration is an important factor to be able to interact with one or more of the olfactory receptors [55].

Aroma compounds constitute one of the most important segments in the perfumery, food, and cosmetic industries. Hence, the global flavors and fragrances market size reached a high peak in 2019 with an estimated scale of USD 28,193.1 million, and it is projected to reach USD 35,914.3 million by 2027 with an estimated compound annual growth rate (CAGR) of 4.7% [56].

In general, the production of aroma compounds can be carried out via chemical synthesis, extracted directly from a natural matrix or derived from biotechnological processes [57]. However, research leans towards the fermentation process using microorganisms because it is an economic alternative to the difficult and expensive extraction from raw materials such as plants and avoids the environmental impact of the chemical process [58], in addition to consumer preferences for products labeled as “natural” [59].

Studies have shown that solid-state fermentation is a promising process for aroma compound production. Due to its capacity to provide a complex aroma profile, bread waste can be used as an excellent feedstock for the fermentation process without the need for extra nutrients [60,61].

In previous research, Daigle et al. used *Geotrichum candidum* ATCC 62,217 to study the production of fruity fragrance compounds (pineapple-like) on fermented waste bread (35% white breadcrumb and 65% water) [58]. Because of this strain’s affinity for aromatic compounds, it was able to convert the organic acids in bread into fatty acids esters, primarily ethyl esters of acetic, propionic, butyric, and isobutyric acids. The fermentation temperature, agitation, and time were only a few of the parameters that the authors optimized. Thus, after 48 h at 30 °C, the produced volatile chemicals reached their optimal levels.

Terpenes, the largest class of naturally occurring aromatic hydrocarbons that serve as a foundation for smell and flavor, can also be produced from bread waste. This strategy was examined in a recent work by Styles et al. [62] who used thermophilic bacteria in a metabolic engineering project. In this work, bread waste was employed as a feedstock for *Parageobacillus thermoglucosidasius* NCIMB 11,955 due to its high starch content (80%). This strain can directly break down starch to liberate maltose, which can be catabolized as a carbon source, characterized by starch-degrading enzymes in its genome, including -amylase and neopullulanase. After 48 h of growth and at a temperature of 55 °C, the highest production yield of terpenes was noted in a 2% bread waste media (*w*/*v*) versus maltose media (2% (*w*/*v*)). However, this production pertains to a laboratory scale and needs further experiments to convert it into industrial production [62].

Prior research has thoroughly investigated other food waste for aroma compound synthesis. Aggelopoulos et al. [60] appraised the formation of high amounts of Ɛ-pinene (which has chemical and physical properties similar to α-pinene and β-pinene) using food waste mixtures (orange and potato pulp) via solid-state fermentation. For this fermentation process, they used a natural mixed dairy culture consisting of symbiotic consortia of yeasts and bacteria embedded in a polysaccharide matrix (Kafir), which yielded an estimated production rate of 4 kg/ton of the treated substrate.

### 4.10. Bread Waste for Enzyme Production

An enzyme is a protein molecule that is used as a catalyst for chemical reactions. Hence, it has become a commodity chemical for various industrial products, and it is used in a variety of applications, such as in detergents, textiles, food processing, animal feed, leather, paper, and chemicals production [63]. Due to the increasing demand for enzymes in industries such as food and beverages, biofuels, animal feed, and household cleaning, the global enzymes market reached a high peak in 2018 with an approximate scale of USD 9.3 billion. Furthermore, it was estimated to grow even more from 2019 to 2025 with an annual growth rate of 7.1% [64]. However, a major problem in the industrial application of enzymes is the cost associated with their production.

Recently, the concept of waste valorization has sparked a worldwide debate encouraging researchers to look for low-cost resources for enzyme production. As such, bread waste provides an excellent fermentation feedstock for enzyme synthesis due to its availability year-round and its richness with high carbohydrate content and additional nutrients [65].

In general, enzyme production can be carried out via solid-state fermentation by using bread waste as a substrate for microorganisms’ reactions. As described in the study of Benabda et al. [66], a process was conducted to co-produce two industrial enzymes, α-amylase and protease, via *Rhizopus oryzae* on humidified bread waste. The solid-state fermentation process was carried out in sterile aluminum trays while pH and humidity were adjusted, respectively, to 5.5 and 65%. Interestingly, after 120 h, the results yielded high levels of α-amylase (100 U/g) and protease (2400 U/g) production [66].

In a previous study, Melikoglu et al. [4] also used bread waste in a solid-state fermentation process, via *Aspergillus awamori,* to produce glucoamylase and protease enzymes. The optimal fermentation conditions for enzyme production were established as a 20 mm bread particle size, 67% moisture, and a duration of 144 h. Under these conditions, maximum yields of glucoamylase and protease production that reached up to 114.0 and 83.2 U/g bread were obtained, respectively [4]. Another study by Haque et al. [49] investigated the production of the same enzymes using bakery waste. The solid-state fermentation process via *Monascus purpureus* yielded the highest glucoamylase and protease activities of 8 U/g and 117 U/g, respectively, with an initial moisture content of 55% and 65%, respectively [49]. Table 2 lists the enzymes and their production processes from bread waste.

### 4.11. Bread Waste as a Growth Medium

Bread is a highly nutritious food, and that is not only for humans. Thanks to its baking process and its composition, bread is considered a highly digestible food. This property, however, can make it less stable and more susceptible to microbial attack. Typically, 100 g of white bread contains around 50 g of carbohydrates, 37 g of water, and about 8 g of protein. This composition makes bread an excellent and a near-complete source of nutrients for many microorganisms. In fact, in the natural state, the spoilage of bread consists usually of a solid-state fermentation involving filamentous fungi, often species of *Aspergilli* [4]. Hence, this natural fermentation process has triggered the interest of researchers to study bread waste as a growth medium for the culture of microorganisms used to produce potentially valuable products.

Fundamentally, bread waste can be used as a growth medium in two different ways.

Firstly, raw bread waste can provide an excellent source of glucose for many microorganisms. As described in the study of Thyagarajan et al. [67], a breadcrumb substrate was tested as a potential carbon source for marine microorganisms, *Thraustochytrium* sp. AH-2 and *Schizochytrium* sp. SR21, for lipid production via static fermentation conditions [67].

Moreover, bread waste can be processed before using it as growth media. To achieve this, bread waste goes through a hydrolysis step to break down the macromolecules (i.e., starch and protein) into monomers (i.e., glucose and free amino nitrogen) via proteolytic and amylolytic enzymes. The applicability of this process was tested by Benabda et al. [68] through the hydrolysis of white baguette waste using enzyme preparations to obtain two different growth media, Medium I and II, which contained a mixture of alpha-amylase and amyloglucosidase (Medium I) and a mixture of alpha-amylase, amyloglucosidase, and protease enzymes (Medium II). Then, they proceeded by testing those media on *Saccharomyces cerevisiae* culture. After 48 h of fermentation, important biomass production was recorded in media (I) and (II) (respectively, 1.04 × 10^8^ CFU/mL and 2.74 × 10^8^ CFU/mL) compared to the control medium (1.63 × 10^8^ CFU/mL) [68].

In a recent study, Verni et al. [69] also tested bread waste hydrolysate on optimizing media for the growth of lactic acid bacteria, yeasts, or fungi as a starter for the dairy, bakery, and wine industries. Interestingly, their results showed that the growth yeasts and fungi exceeded the reference media commonly used for their cultivation [69]. This shows that bread waste can be highly considered as a realistic option for valorization and re-use as a growth medium.

### 4.12. Other Methods for Bread Waste Exploitation

Bread waste represents a substantial rich source that can be further exploited for more advanced applications. Hence, much research has been conducted recently to produce high-value products, including chemicals (i.e., 2-keto-d-gluconic acid acetoin and d-2,3-butanediol), materials (i.e., bioactive porous scaffolds, graphene sheets, polyhydroxybutyrate), and nutrient-enriched products. A summary of the various approaches adopted by researchers working on these applications is given in Table 3.

## 5. Logistical Challenges of Utilizing Bread Waste as a Feedstock

As described in the previous sections, bread waste is an excellent economical feedstock to produce valuable chemicals in various industries. In fact, utilizing bread waste can drive the costs down of the production process alongside its remarkable environmental impact. However, the constant use of bread waste as a feedstock presents challenges. 

To produce fermentation-based chemicals such as biohydrogen, succinic acid, and aroma, waste materials must be quickly transported to the site of fermentation to prevent the risks of decaying and becoming an environmental health hazard as well as degrading the feedstock to an extent that it is no longer usable. Therefore, the proper preparations and equipment need to be ready for delivering tons of bread waste, especially at a commercial scale, within a short period to be ready for pre-processing and eventual fermentation. Moreover, collecting bread waste from bakeries and supermarkets enables clean flow; however, it is not the case for domestic collection. At the domestic level, the contamination of bread waste with other food wastes is inevitable, presenting a major obstacle as the separation process can be quite challenging. In addition, being mixed with other types of food waste can accelerate its degradation. To prevent the challenges of domestic collection, regulation of the quality of waste material and transportation timings must be carefully controlled, which requires additional investments. 

While using bread waste might significantly reduce GHG emissions, processing the waste runs the risk of having an adverse effect on the environment. In fact, the additional transportation required for the collecting of bread waste may contribute to a further increase in GHG emissions. In this situation, it is important to pay close attention to the degrees of transportation needed for the various processing routes. Additionally, it is important to monitor the fermentation process when using bread waste as a feedstock because some processes, such as the creation of ethanol, might produce CO_2_ as a byproduct. Therefore, proper measurements should be made to swiftly and thoroughly extract and contain fermentative CO_2_ [7].

Another challenge posed by bread waste exploitation is the economic slowdown that the world faced in the past 3 years due to the pandemic. In fact, to ensure economic feasibility, the design of the manufacturing process must consider capital and operational costs, therefore requiring consumer and governmental support [77], which is available only in places where there is significant policy support for innovative waste management technologies to implement the production of bread-waste-based chemicals on an industrial scale. For example, Renewable Transport Fuel Certificates (RTFCs) in the UK provide sufficient financial support for organic-waste-derived biofuels and make industries, such as ethanol production from municipal solid waste, financially viable [78].

The geopolitical tension caused by the Russia–Ukraine war is also a major economic factor to consider since wheat scarcity and the surge in prices keep destabilizing food security [79]. Any reliant production could be in danger, and investment projects could be constrained due to the decrease in bread output, and consequently, bread waste.

One final word of caution when using bread waste as a feedstock would be to consider the quantities required, especially with scale-up to the commercial level. Although it may seem promising at the laboratory level, using bread waste for chemical production at a commercial scale will inevitably present numerous challenges in many areas, such as the amounts of energy required, the quantities of expensive reagents used in the chemical reaction such as for HMF production, and the amount of feedstocks needed. In fact, between beer and ethanol production, for other industries to fully rely on bread waste as a feedstock will inevitably present a huge obstacle to providing consistent productivity.

## 6. Conclusions

In order to create a more sustainable society in the upcoming years, the United Nations has prioritized reducing food waste. The production of high-value-added products can use the vast amounts of food waste that are produced worldwide as a bioresource.

Bread is no exception for food waste. Considering the great amounts that are being piled up annually, waste management should be the main focus in order to minimize bread waste as much as possible, especially in the post-pandemic years during which the price of cereal and shipping costs have increased drastically. However, if waste cannot be further reduced, it would be imperative to resort to possible solutions and methods for repurposing bread surplus. In fact, bread waste represents an interesting raw material due to its rich composition, easy conversion to glucose, availability, safety, and costlessness.

In this regard, many papers have demonstrated the feasibility of transforming bread waste into glucose and fructose syrups, organic acids, pigments, proteins, ethanol, and biohydrogen. Moreover, scale-up investigations should be considered seriously in order to evaluate the industrial feasibility of the proposed technology. In fact, the discussed yields of each approach provide significant insights into the scale-up option at the industrial level and the potential founding of industries based on bread waste valorization. In this framework, these industries will definitely provide strategic road-mapping activities aimed at strengthening the circular economy.

Finally, it can be suggested that the use of waste bread to produce added-value products should be seriously considered by local governments as part of their strategy in an attempt to produce eco-friendly materials and bioenergy.

## Figures and Tables

**Figure 1 molecules-27-08410-f001:**
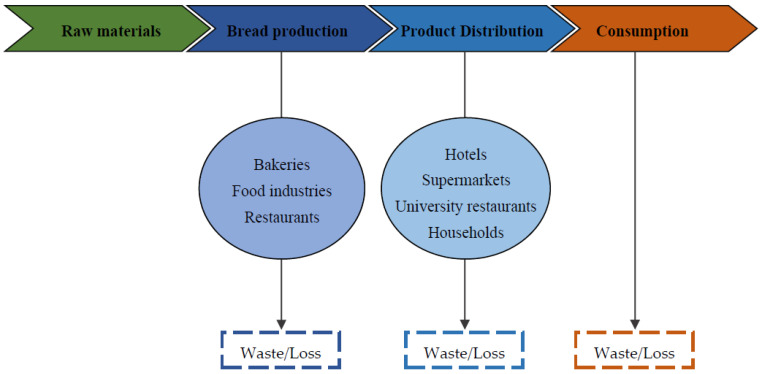
Bread life cycle.

**Figure 2 molecules-27-08410-f002:**
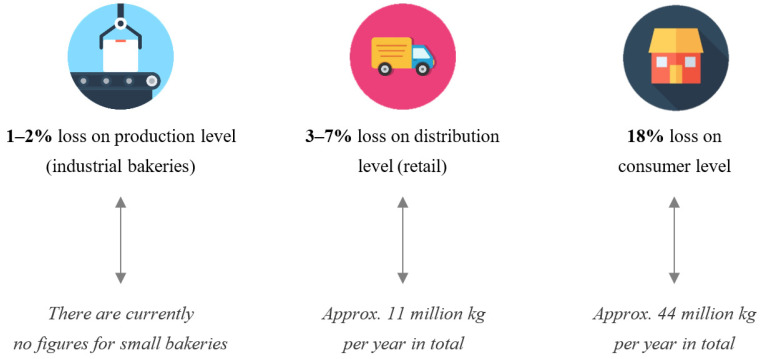
Bread waste estimation in each phase of the supply chain in Flanders [15].

**Figure 3 molecules-27-08410-f003:**
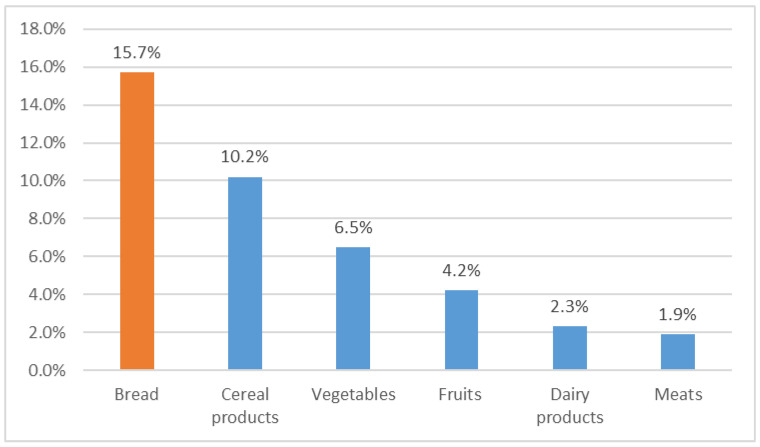
Approximate share of food waste (%) [18].

**Table 1 molecules-27-08410-t001:** Nutritional content and chemical composition of bread.

Property	Quantity	Ref.
Embodied Carbon (kgCO_2_e/t)	1400	[12]
Energy (kcal/kg)	2740	
Starch (g/slice)	45.34	[4]
Protein (g/kg)	106.7	[12]
Kjedahl Nitrogen (g/kg)	17.07	
Phosphorous (g/kg)	1.29	
Potassium (g/kg)	14.1	
Water (%)	35	
Ash (g/slice)	2.26	[4]
Total Solids, TS (%)	65	[12]
Volatile Solids/Total Solids, VS/TS	0.87	
Specific methane potential (m^3^ CH_4_/t VS)	350	

**Table 2 molecules-27-08410-t002:** List of enzymes produced from bread waste substrate via solid-state fermentation.

Enzymes	Microorganisms	Fermentation Conditions	Reference
α-Amylase	*Rhizopus oryzae*	pH = 5.5 Humidity = 65% Temperature = 30 °C Incubation time = 120 h	[66]
Protease	*Rhizopus oryzae*	pH = 5.5 Humidity = 65% Temperature = 30 °C Incubation time = 120 h	[66]
	*Aspergillus awamori*	Humidity = 67% Temperature = 30 °C Incubation time = 144 h Bread particle size = 20 mm	[4]
	*Monascus purpureus*	Humidity = 65% Temperature = 30 °C Incubation time = 72 h	[49]
Glucoamylase	*Aspergillus awamori*	Humidity = 67% Temperature = 30 °C Incubation time = 144 h Bread particle size = 20 mm	[4]
	*Monascus purpureus*	Humidity = 55% Temperature = 30 °C Incubation time = 72 h	[49]

**Table 3 molecules-27-08410-t003:** Summary of further applications for the utilization of wasted bread and bakery products.

Application	End Product	Method Used	Reference
Tissue engineering	Bread-derived bioactive porous scaffolds	Use of bread waste in fabrication of a macroporous template to produce bone tissue engineering bioactive glass-derived scaffolds	[70]
Material production	Graphene sheets	Use of ground bread waste and deionized water via hydrothermal method to produce single-layer and few-layer graphene sheets	[71]
Fermentative production	Polyhydroxybutyrate (PHB)	Batch and fed-batch cultures using bakery waste hydrolysate as nutrient source, with seawater, for *Halomonas boliviensis*	[72]
	2-Keto-d-gluconic acid	Use of waste bread hydrolysate as a substrate for *Pseudomonas reptilivora* NRRL B-6 fermentation process	[73]
	Recycled baking ingredient	Production of bread slurry rich with dextran using in situ exopolysaccharide production by *Weissella confusa* A16	[74]
	Protein-enriched food product	Stale sourdough bread is used as a substrate for SSF by the edible filamentous fungi *Neurospora intermedia* and *Rhizopusoryzae*	[75]
	Acetoin and d-2,3-butanediol	Bakery waste hydrolysate used in batch fermentation by *Bacillus amyloliquefaciens*	[76]

## Data Availability

This review did not generate any new data.

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
