# Peer review of "Bread Surplus: A Cumulative Waste or a Staple Material for High-Value Products?"

_molecules, 2022, doi:10.3390/molecules27238410_

Round 1

Reviewer 1 Report

This manuscript is about the re-use and valorisation of bread waste. The topic is meaningful and interesting. The review has systematically summarized current approaches and related research progresses in this aspect.

The manuscript is generally well organized and written, and references are appropriately cited. It can provide useful information for readers who are interested in this field.

Other comments:

Table 1 -- These index are usually in ranges. Are these figures are averages? 

The last line in Table 1 -- the “3” in m3 should be a superscript.

Figure 3 -- “bead” should be “bread”?

In-text citations --“and al.” should be “et al.”

Author Response

Reviewer 1

Authors’ response: We appreciate the Reviewer’s input to review the manuscript and give this positive comment.

  1. Table 1 -- These index are usually in ranges. Are these figures are averages?

Authors’ response: we appreciate the reviewer’s observation and thank you for this opportunity to clarify this point. Yes, according to the references cited in table 1. The gas emission (methane and CO2) values were determined using the average value of gas generated multiplied by the emissions factors detailed elsewhere (Carlsson & Uldal, 2009; Hoolohan et al., 2013). As for the other indexes, they represent the average values taken from the food composition database provided by reports from USDA (2013, 2017) and FAO (2016).

  1. The last line in Table 1 -- the “3” in m3 should be a superscript.

Authors’ response: Thank you for pointing this out. This error has been corrected in the last line in table 1 figured on page 33

  1. Figure 3 -- “bead” should be “bread”?

Authors’ response: Thank you for pointing this out. The word ‘’bread’’ has been corrected in figure 3 on page 32

  1. In-text citations --“and al.” should be “et al.”

Authors’ response: Thank you for pointing this out. We have revised the manuscript thoroughly and all the in-text citations were changed accordingly

Reviewer 2 Report

In the manuscript molecules-2036533, the authors aimed to investigate the highlights of potential applications for recycling bread waste into valuable products through chemical and biotechnological approaches.

I have some doubts and suggestions after reading manuscript:

1.       There are some spelling and grammatical mistakes. I suggest inviting a native English speaker to revise the whole text.

2.       In the introduction, please explain what type of bread waste is recycled as animal feedstock. Maybe you think it is evident, but it isn’t, especially since you are writing about the prospective animal feedstuff.

3.       I think that you should cite in the discussion the lactic acid production for potential animal feedstuffs, as you have mentioned in lines 271-275. Have a look at these updated papers:

https://doi.org/10.3390/fermentation8060248

https://doi.org/10.3389/fchem.2022.823005

https://doi.org/10.3389/fvets.2022.896270

4. Please add some recommendations for practice to your conclusions. Scientifically, it is a good job, but what does it mean in practice?

Author Response

Reviewer 2

Authors’ response: We appreciate the Reviewer’s input to review the manuscript and give these insightful suggestions.

  1. There are some spelling and grammatical mistakes. I suggest inviting a native English speaker to revise the whole text.

Authors’ response: we apologize for our lack of revision. We have now edited the whole manuscript by a native English speaker as suggested.

  1. In the introduction, please explain what type of bread waste is recycled as animal feedstock. Maybe you think it is evident, but it isn’t, especially since you are writing about the prospective animal feedstuff.

Authors’ response: we apologize for our lack of information. Indeed, the introduction part has been updated, the suggested content has been added to the manuscript on page number 4:

‘’ To comply with environmental quality objectives, set by waste management systems, solutions for limiting bread waste have progressively been established. One of the valorization routes is using bread waste as animal feed. In fact, feeding animals food scraps is one of the bases of the Food Recovery Hierarchy, and Farmers have been doing this for centuries. By following legislation for the selection of proper and safe food waste, bread leftovers or stale dry bread offers a great alternative method to obtain animal feed at a much lower cost (US EPA, 2022).’’

  1. I think that you should cite in the discussion the lactic acid production for potential animal feedstuffs, as you have mentioned in lines 271-275. Have a look at these updated papers:

https://doi.org/10.3390/fermentation8060248

https://doi.org/10.3389/fchem.2022.823005

https://doi.org/10.3389/fvets.2022.896270

Authors’ response: Thank you for suggesting. The second reference ‘’https://doi.org/10.3389/fchem.2022.823005’’ has been added in part ‘’ 4.3 Lactic acid production’’, on pages 9-10. As for the other two references, we agree that this is an important consideration, and it would have been interesting to explore this aspect. However, in our study, this would not be possible because it is beyond the scope, as the part of lactic acid in this review specifically assess the production methods of lactic acid from bread waste.

  1. Please add some recommendations for practice to your conclusions. Scientifically, it is a good job, but what does it mean in practice?

Authors’ response: Thank you for pointing this out. We think this is an excellent suggestion. Accordingly, the suggested content has been added to the manuscript in the conclusion part, on page number 20:

‘’ In fact, the discussed yields of each approach provide significant insight into the scale-up option at industrial level, and the potential founding of industries based on bread waste valorization. These industries will definitely provide in this framework strategic road-mapping activities aimed at strengthening the circular economy.’’

Reviewer 3 Report

The reviewed manuscript (molecules-2036533) is a very interesting, well-written comprehensive, and rich source of information presenting the possible use of bread surplus, which is often simply wasted, through its chemical and biotechnological recycling into valuable products. This review highlights the urgent need to reuse and valorize that wasted bread, given that bread surplus, particularly high in developed countries, is a serious problem with direct and negative environmental impact.
The authors rightly point out that, especially nowadays, the management of post-production food waste is a huge challenge on both governmental/societal levels as well as for technologists/biotechnologists who are looking for the possibility of their management through processing with the use of increasingly improved and more efficient processes and technologies.

I just have a few comments to consider that might further improve this already very good manuscript:
- I suggest adding a short methodological paragraph explaining how the selection of literature was carried out (what databases were searched, based on what keywords, and what was the time range of searching)
- please unify the method of citing references in the text,  "Author et al." or "Author and al.";
-please omit the first names of the cited author, it is common to quote authors by giving their surname

Author Response

Reviewer 3:

Authors’ response: We appreciate the Reviewer’s input to review the manuscript and give this positive comment and these insightful suggestions.

  1. I suggest adding a short methodological paragraph explaining how the selection of literature was carried out (what databases were searched, based on what keywords, and what was the time range of searching)

Authors’ response: We think this is an excellent suggestion. We agree with the reviewer’s assessment. Accordingly, a short methodological paragraph has been added after the introduction part on page number 4:

‘’ Extensive bibliographic research was conducted using the scientific databases Web of Science, PubMed, Scopus, and annual reports from organizations (FAO, UNEP) by selecting articles and reviews from the past decades (2008–2022). The following keywords were used in the research: technology, valorization, value-added products, stale bread, surplus bread, and bread waste. To facilitate the screening, the research was refined by using the terms: bread making, ethanol, lactic acid, succinic acid, biohydrogen, Hydroxymethylfurfural, proteins, aroma compounds, growth medium, and enzymes.

All the articles that met the criteria to be included in the study, were analyzed. Reports obtained were evaluated by screening abstracts to discard unnecessary, incomplete, or irrelevant literature. A total of 80 articles were analyzed in full text and classified by groups based on the molecules studied, types of technologies used, and production yields.’’

  1. please unify the method of citing references in the text, "Author et al." or "Author and al.”.

Authors’ response: Thank you for pointing this out. We have revised the manuscript thoroughly and all the in-text citations were unified as "Author et al.".

  1. please omit the first names of the cited author, it is common to quote authors by giving their surname

Authors’ response: Thank you for pointing this out. We have revised the manuscript thoroughly and all the first names of the cited authors have been omitted.

Round 2

Reviewer 2 Report

Dear authors

I notice your effort to improve the paper. I suggest this current form is considered adequate for publication; however, those figures especially figure 3, needs improvement, better selection to use the color between bread vs other.

Best regards